



# Simulating sub-hourly rainfall data for current and future periods using two statistical disaggregation models - case studies from Germany and South Korea

Ivan Vorobevskii[1], Jeongha Park[2], Dongkyun Kim[2], Klemens Barfus[1], Rico Kronenberg[1]

[1] Faculty of Environmental Sciences, Department of Hydrosciences, Institute of Hydrology and Meteorology, Chair of Meteorology, Technische Universität Dresden, Tharandt, 01737, Germany

[2] Department of Civil and Environmental Engineering, Hongik University, Seoul, 04066, Korea

*Correspondence to: Ivan Vorobevskii (ivan.vorobevskii@tu-dresden.de) & Jeongha Park (parkjungha1121@gmail.com)*

**Abstract**. Simulation of fast reacting hydrological systems often requires sub-hourly precipitation data to develop appropriate climate adaptation strategies and tools, i.e. upgrading drainage systems and reducing flood risks. However, this sub-hourly data is typically not provided by measurements and atmospheric models, and many statistical disaggregation tools are applicable only up to an hourly resolution.

Here two different models for disaggregation of precipitation data from daily to sub-hourly scale are presented. The first one is a stochastic disaggregation model based on first-order Markov Chains and Copulas (WayDown), while the second one is a stochastic precipitation generator based on a double Poisson process (LetItRain). Both approaches aim to reproduce observed precipitation statistics over different time scales.

The developed models were validated using 10-min radar data representing 10 climate stations in Germany and South Korea, thus covering various climate zones and precipitation systems. Various statistics were compared including mean, variance, autocorrelation, transition probabilities, and proportion of wet period. Additionally, extremes were examined, including the frequencies of different thresholds, extreme quantiles and annual maxima. While both models successfully reproduced the observed statistics, WayDown was better than LetItRain at reproducing the ensemble median showing strength in precisely refining the coarse input data. In the meantime, LetItRain produced rainfall with greater ensemble variability capturing a variety of scenarios that may happen in reality. Both methods reproduced extremes in a similar manner: overestimation until a certain threshold of rainfall thereafter underestimation.

Finally, the models were applied to climate projection data. The change factors for various statistics and extremes were computed and compared between historical (radar) and the climate projections on a daily and 10-min scale. Both methods showed similar results for the respective stations and RCP scenarios. Several consistent trends jointly confirmed by disaggregated and daily data were found for mean, variance, autocorrelation and proportion of wet periods. Further, they





presented similar behavior of annual maxima for the majority of the stations for both RCP scenarios in comparison to the daily scale, namely a similar systematic underestimation.

**Introduction**

Urban hydrological systems are characterised by large impervious surface areas and dense underground drainage networks, so their response to rainfall is direct and fast (Meierdiercks et al., 2010; Sohn et al., 2020). In such systems rainfall events

with different fine-scale temporal variability may lead to significant different patterns of flooding (Oh et al., 2016; Dao et al., 2020a, b, 2022; Park et al., 2021) and the associated disasters such as landslides, degradation of water quality and ecosystem, and risks to public health and safety. Therefore, the acquisition of fine-scale rainfall data is critical for accurate estimations, understanding of the causes and impacts of floods (Berne et al., 2004; Vorobevskii et al., 2020), thus enabling the design of sustainable and resilient urban drainage systems that can adapt to changing climate conditions.

However, fine-scale (e.g. 10-minutes or finer) rainfall data suitable for urban flood analysis are often unavailable. In-situ gauge data are usually measured at hourly or daily intervals due to the issues of initial cost, maintenance, data quality and applicability, or time series are still not long enough for reliable statistics. In addition, most future rainfall projection data produced by recent global (e.g. daily) and regional (e.g. hourly) climate models for downscaling have coarse temporal resolution (Dyrrdal et al., 2018; Iles et al., 2020) making it difficult to precisely analyse urban flood risks associated with

climate change. Although recent climate models allowing for explicit modelling of deep  convection (Prein et al., 2015) can simulate 5 minute precipitation fields (Meredith et al., 2020), these products will not be available on a large scale in the near future due to computational and data storage limitations (Schär et al., 2020).

The lack of fine-scale precipitation data can be tackled by rainfall downscaling techniques (Maraun et al., 2010). Two main approaches can be distinguished: disaggregation models and stochastic generation models. Rainfall disaggregation models

(Müller-Thomy et al., 2018) refine the temporal resolution of the original rainfall time series. Therefore, the sum of the disaggregated fine-scale rainfall contained in the original coarse time step is similar to (canonical models) or precisely the same as (micro-canonical models) the original coarse rainfall value. Stochastic generation models on the other side aim to reproduce statistics of the original rainfall time series. They employ techniques such as linear regression, probability density function fitting, and machine learning to characterise rainfall processes (e.g. event depth, event duration, inter-event time,

etc.), based on which variables comprising rainfall processes are produced and superposed on an empty time axis to synthesise fine-scale rainfall time series. Therefore, stochastic generation models do not preserve the rainfall records of the original coarse data, but can generate an infinite length of synthetic time series, thus are mainly used as the input data of the uncertainty analysis of disaster risks based on the Monte-Carlo simulation.

While many rainfall disaggregation (Koutsoyiannis and Onof, 2001; Kossieris et al., 2018; Lombardo et al., 2017; Müller

and Haberlandt, 2015; Müller-Thomy, 2020) and stochastic generation models (De Luca and Petroselli, 2021; Fatichi et al.,





2011; Papalexiou, 2018; Pidoto and Haberlandt, 2023; Peleg et al., 2017; Semenov and Barrow, 1997; Verdin et al., 2018) have been developed, only very few studies address models that can produce fine-scale (e.g. finer than 30-minutes) rainfall data. Licznar et al., 2011 developed a disaggregation-type model based on a random cascade. They introduced a unique scale-dependent cascade coefficient to produce fine-scale (5-minute) to improve the performance in reproducing fine-scale rainfall depth distribution as well as key statistics such as the mean and standard deviation of the annual rainfall maxima. Lombardo et al., 2017 proposed a disaggregation-type model that simulates rainfall time series with given dependence structure, wet/dry probability, and marginal distribution at a finer time scale, preserving full consistency with variables at a parent coarser time scale. The suggested model was tested on 30-minute rainfall data of Viterbo, Italy and accurately reproduced the marginal distributions of the characteristic variables of both fine and coarser scale rainfall time series as well as correlation structure, intermittency, and clustering. The disaggregation model of Kossieris et al., 2018 combines Bartlett-Lewis process to generate rainfall events along with adjusting procedures to modify the low resolution variables (i.e. hourly) so as to be consistent with the high-resolution ones (i.e. sub-hourly). The suggested model successfully replicated important statistical properties up to 5-min scale at a wide range of time scales including improved fit for intensity-duration dependent internal rainfall structure, skewness, extremes, and dry proportions compared to its predecessor (Koutsoyiannis and Onof, 2001). Another disaggregation model of Müller and Haberlandt, 2018 combined tri-furcation and bifurcation random cascade method to obtain the 5-minute rainfall time series. The method was tested on the 24 gauges of Lower Saxony of Germany and showed an improved performance in reproducing regular statistics and extremes as well as sewage system behavior compared to the conventional bifurcation-based cascade models. (Park et al., 2021) suggested a stochastic generator-type model that is also based on the Bartlett-Lewis process. They modified the original model structure that assumes the rectangular shape of the rain cell to the sinusoidal shape so the model can produce 5-minute rainfall given hourly rainfall. The 5-minute rainfall synthesised by the modified model contained more realistic extreme rainfall values as well as flooding behaviour in urban environments compared to the model assuming rectangular rain cell structure.

In spite of this progressive evolution of the fine-scale rainfall downscaling models, most studies have focused on development of a single model or on validating existing models of the same kind at multiple study sites (Kim et al., 2013, 2016a, 2017; Takhellambam et al., 2022; Wang et al., 2021). Only a few studies performed comparative analysis of multiple disaggregation models as this study. Pui et al., 2012 compare three typical types of rainfall downscaling models (random multiplicative cascade model, point process model, and resampling model) that disaggregates daily rainfall to hourly resolution. The comparison was performed at four point locations in Australia with different climatic regimes. They discovered that all the models are found to simulate reasonably well the commonly used statistical measures of rainfall at the hourly time step while the microcanonical cascade model overestimated the hourly rainfall variance; and that extreme rainfall values were under- or over-estimated by the cascade models. However, no studies have yet tested disaggregation models of different types at the fine timescale (10-minutes) critical for urban system analysis as well as for a variety of rainfall characteristics under different climatic systems.





This study aims to utilise promising techniques for temporal fine-scale downscaling of future rainfall by applying models,
which do not have many statistical requirements regarding correlations of process driving variables. Hence which are not
depending on many input datasets. Furthermore the model application should be simple, mostly automatically and fast.

In this view, the first aim of this study is to compare newly developed disaggregation-type model and a stochastic
generation-type model for the task of fine-resolution rainfall downscaling. The disaggregation-type model of this study is
composed of a unique new combination of Markov Chain for simulating binary sub-daily events alignment and copula-based
sampling of actual precipitation values, which to our knowledge has not been attempted in our field yet. The stochastic
generation-type model of this study is an advanced version of the Poisson cluster rainfall generation model (Kaczmarska et
al., 2014; Kim and Onof, 2020). It can synthesise future rainfall under climate change given a change factor (ratio of mean of
future to current rainfall) that can be easily obtained from climate change rainfall products. The model has a unique structure
to obtain parameters for future rainfall generation that to our knowledge has not been tried by other studies. In addition, two
focus regions of this study covers a wide range of climate and rainfall characteristics: while both Germany and Korea have
temperate climates, Germany has a more moderate and stable rainfall pattern with less regional variation (600-1800
mm/year), while Korea is characterised by more regional variation in its rainfall patterns and amounts (1200-2000 mm/year)
and is subject to long-lasting heavy frontal rainfall during the summer months and intense typhoons. Therefore, the
validation of these models for this variety of rainfall characteristics should reveal the suitability and limitations of model
application in general and provides some insight into transferability to other regions.

One of the novelties of this study is due to the use of radar rainfall data instead of gauge data. As opposed to gauges that can
accurately observe rainfall depth at a point location, weather radar observes rainfall in a fine granular format over a wide
spatial coverage. Thus, the radar data could be more suitable in understanding regional climate and its non-stationarity as
well as in the applications of the climate projection data. In addition, the chronic issue of radar rainfall measurement
accuracy has been constantly addressed through various methods of the Z-R relationship improvement (Alfieri et al., 2010;
Kim et al., 2021; Kirsch et al., 2019) and radar-gauge merging (Goudenhoofdt and Delobbe, 2009; Han et al., 2021; Ochoa-
Rodriguez et al., 2019; Sinclair and Pegram, 2005). Consequently radar rainfall products are being actively adopted by many
studies on understanding of hydrologic systems (Ghimire et al., 2022; Wijayarathne et al., 2020, 2021) as well as operational
flood warning systems (Ramly et al., 2020; Liu et al., 2021). However, to our knowledge, there is only one study of Jasper-
Tönnies et al., 2012 who applied 5 min radar data and used a relatively simple methodic of 'objective weather types' to pick
an observed radar event with similar daily sum to downscale climate projection data. No other studies have investigated the
applicability of the recent radar data with improved quality (Park et al., 2014; Winterrath et al., 2018) for rainfall
downscaling, especially based on two unique methods with contrasting traits. Further aim of the study is to test the described
models on climate projection data. Therefore, after being calibrated and validated for the current period, both models were
employed to produce 5-minute rainfall data corresponding to the RCP scenarios of 2.6 and 8.5. The fine-scale data produced
by each of the methods were compared in terms of change factors of various statistics and annual maximum rainfall values.



The research questions discussed in this article are as follows: (1) How well do two different fine-scale rainfall downscaling models produce data? Are they suitable for reproducing important rainfall statistics as well as extreme values? (2) What are the differences and similarities between the presented types of downscaling models? (3) How might future fine-scale rainfall change according to the respective models?

The paper is organised as follows: Section 2 provides a detailed description of the models, study areas, and data sources, including both radar data and future projection data. Section 3 presents the results of our analysis, including a comparison of the two models and an assessment of their accuracy. Finally, in Section 4, we summarise our findings and discuss their implications for future research and revenue opportunities in the field of rainfall disaggregation.

## Methods and Data

### WayDown

WayDown is an automated precipitation disaggregation model, which was developed at the Chair of Meteorology of Technische Universität Dresden. It is wrapped in an R-package and is available on GitHub (https://github.com/hydrovorobey/WayDown) along with a test dataset.

The principle scheme is presented in Figure 1. WayDown disaggregates daily precipitation by iterating over each day and keeping the daily sums consistent. This could be i.e. station or climate projection datasets. In the first step, a precipitation event is selected using the high-resolution reference data for the desired month. For that, all reference events with daily sums similar to the input value are subsetted. The Markov Chain's two-state transition matrix is estimated from these events, which is then used to sample binary 5-minute precipitation time-series for a given day. Thereafter, the actual precipitation heights are sampled for respective intervals with binary precipitation, which can be either single or consecutive sub events. The first value for the sub-event is directly sampled (with empirical probability weights) from the observed data accounting for a daily value and the respective month of occurrence. The second and following time steps with precipitation are selected based on a 2D empirical beta copula (Segers et al., 2017) constructed from the reference high-resolution data set in a way that the subsequent precipitation height depends on the previous one, thus representing the lag-1 autocorrelation model. Afterwards, the sum of the obtained disaggregated time-series is compared to the original daily input value. If the absolute difference is less than an assumed threshold (i.e. 10%, which was found suitable with regard to computation time for the study sites and might be changed for other datasets) then a proportional correction is applied. Higher values receive a proportionally larger correction than smaller values. For small daily precipitation sums (e.g. less than 1 mm) a higher threshold can be allowed (i.e. 100%) to reduce computational time. Otherwise, the disaggregation process is repeated again until the convergence error value less than threshold is reached: first, the new values are sampled for the same binary time-series and after 30 unsuccessful attempts (default number that can be changed); a new binary time-series for a given day is sampled. Finally, the framework considers day-to-day event transition, taking into account the last precipitation value from





the disaggregated time-series of a previous day to create a consistent event for the current day (Figure 2) which is sampled
from the obtained transition matrix using a starting value of one for a newly created binary time-series.

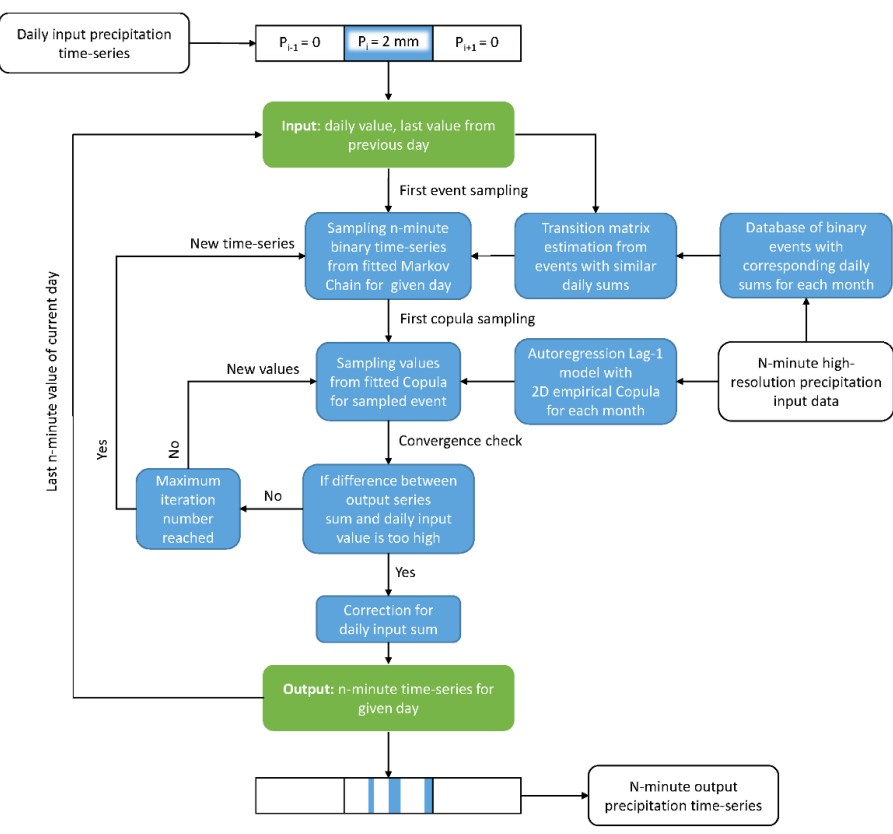


**Figure 1. WayDown framework algorithm**

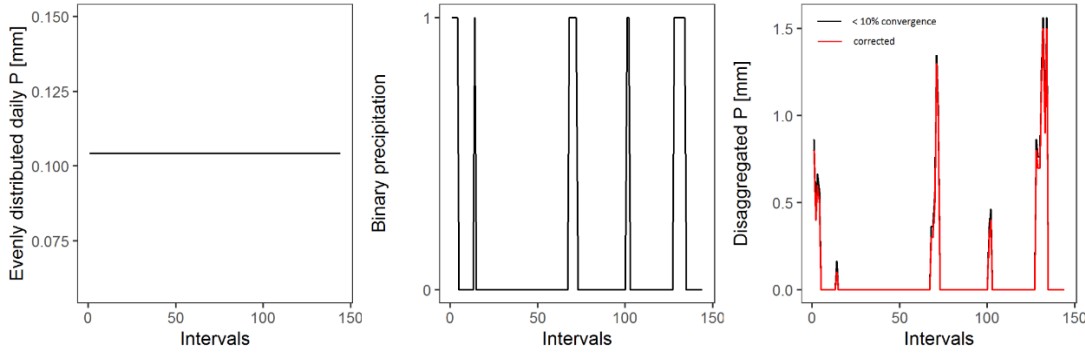

**Figure 2. WayDown disaggregation example for Leipzig (Germany) in July: 10 min resolution radar data (144 intervals), daily
value of 15 mm, last-interval value from yesterday 0.5 mm, obtained convergence error 8%.**

WayDown was tested with the reference input data resolutions from 1, 5, 10 and 30 minutes. The model does not require
substantial resources regarding computational power, time and memory. For example, on a 3.4 GHz 16 GB RAM PC, the





model takes approximately 1 hour to disaggregate 80 years of data to 10 min time-scale using 20 years of reference radar data (see the test dataset with data for Leipzig provided along with the R-package).

**LetItRain**

LetItRain represents a stochastic rainfall generation model and (Kim and Onof, 2020) is an upgraded version of the Poisson cluster model (Kaczmarska et al., 2014; Rodriguez-Iturbe et al., 1988). The model was developed at Hongik University's Hydrology Innovation Laboratory and is available on the lab's webpage (https://sites.google.com/site/hihydrology/projects). LetItRain simulates synthetic rainfall time series based on the assumption that storms arrive following a Poisson process along with different probability distribution functions, which define the duration of storms and the properties (arrival,

duration, and intensity) of rain cells. Existing Poisson cluster rainfall generation models tend to underestimate rainfall extremes, which is why the LetItRain model incorporates the following model improvements. It accounts for the fitting of the first- to third-order moments of observed rainfall. Then, the model inversely relates the rain cell duration to intensity for reproducing short-term extreme rainfall events. Further, LetItRain assumes the gamma distribution for the intensity of the rain cell. Finally, the model applies two shuffling algorithms to reproduce the autocorrelation of storm and long-lasting

rainfall. With these improvements, the model is capable of reproducing observed statistical properties as well as extremes over a wide range of time scales.

The LetItRain application to simulate future sub-hourly rainfall time series is described in Figure 3. Firstly, rainfall statistics for each calendar month are calculated from the high-resolution reference data (i.e. radar rainfall). These statistics include mean, variance, covariance, skewness, and proportion of wet periods for the aggregation interval of 10, 30, 60, 120, 240,

480, and 960 minutes. Here, the whole time series (i.e. including dry periods) are considered. Then, regressions between obtained statistics specified in Figure 4 (upper panel) are derived. One of them is a standard first-order linear regression, which is used for estimating the relationship between mean and variance as well as between mean and proportion of wet periods. The other is the same type of regression, but without intercept, which is used for variances at different aggregation intervals (i.e. 10-minute variance vs. 30-minute variance). The same procedure is repeated for the wet periods of different

aggregation intervals. Secondly, the change factor for the mean value is calculated for a daily scale. It is defined as the ratio of means between historical and future periods and is used to adjust future 10-minute precipitation mean. The change factor approach is e.g. also used by the LARS-WG (Semenov et al., 1998) to generate future daily time series for impact modelling. Thirdly, future rainfall statistics are estimated using the output of previous two steps (Figure 4). The mean value for the future 10-minute rainfall is defined by multiplying the observed 10-minute mean and the change factor. Future mean values

for other aggregation intervals are derived by multiplying 10-minute mean with a fixed factor. Afterwards, variance and proportion of wet periods for the future period are estimated using obtained regressions for historical data and future mean values. Future statistics for covariance and skewness are assigned directly from the reference high-resolution input dataset since here no suitable relationship between rainfall statistics was found and sensitivity of simulations to the variation of those





statistics is weak (Fatichi et al., 2011). Though some indications exist that higher-order moments of precipitation characteristics will change in future (Chan et al., 2016), detailed information from models are missing and thus we adopt higher-order moments from the current climate.

Finally, the model is calibrated to the estimated future rainfall statistics and future sub-hourly rainfall time series are generated. A procedure of model calibration and the calibration results are presented in the Supplementary (Vorobevskii, 2023).

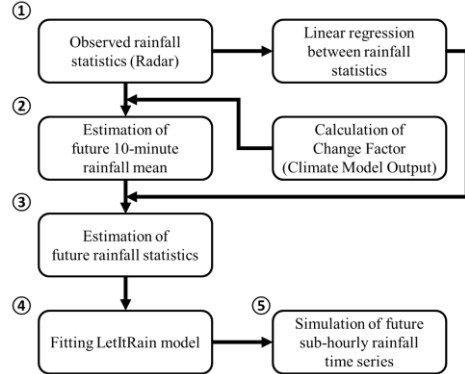

**Figure 3. The procedure for simulation of future sub-hourly rainfall time series using LetItRain model**

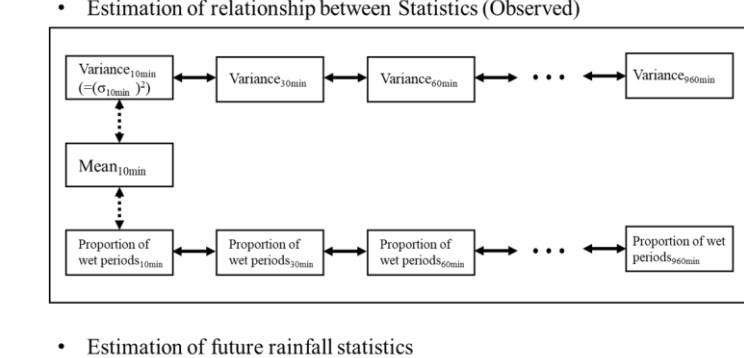

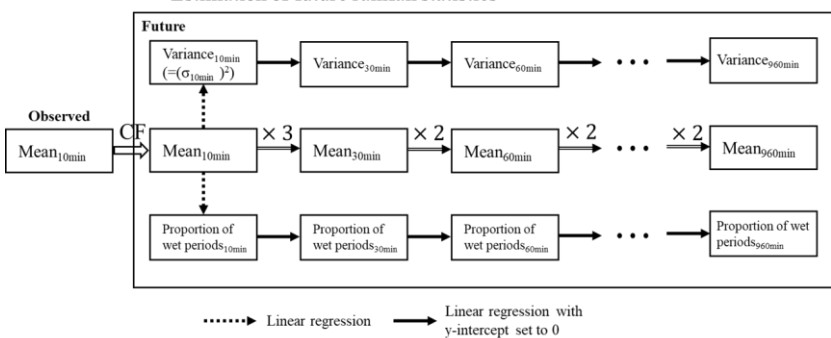

**Figure 4. Rainfall statistics for the estimation of the linear regression relationship (upper part). The procedure for the estimation of future rainfall statistics is shown (lower part).**





### Precipitation data

For the case study, five locations in Germany and five locations in South Korea were chosen (Fig. 5). German sites (Leipzig, Naumburg, Greiz, Hof and Klingenthal) are located in plains and low mountain ranges (middle-eastern part of Germany). They are characterised by continental climate (Dfb - humid continental, warm summer subtype (Kottek et al., 2006)) with annual precipitation sums of around 600-1000 mm. 50-80% of annual precipitation falls in June-September and consists of both convective and stratiform events, while in winter the total amounts are much smaller and are mainly of cyclonic origin (Jung and Schindler, 2019). The Korean sites (Seoul, Gangwon, Daejeon, Gwangju, Busan) are characterised by continental climate (Dwa and Dfa - humid continental, hot summer subtype) with annual precipitation sums of between 1200 and 2000 mm. More than half of this amount falls during the typhoon season (June-September) when a stationary front lingers for about a month in summer. The winter precipitation is typically less than 10% of the annual sums.

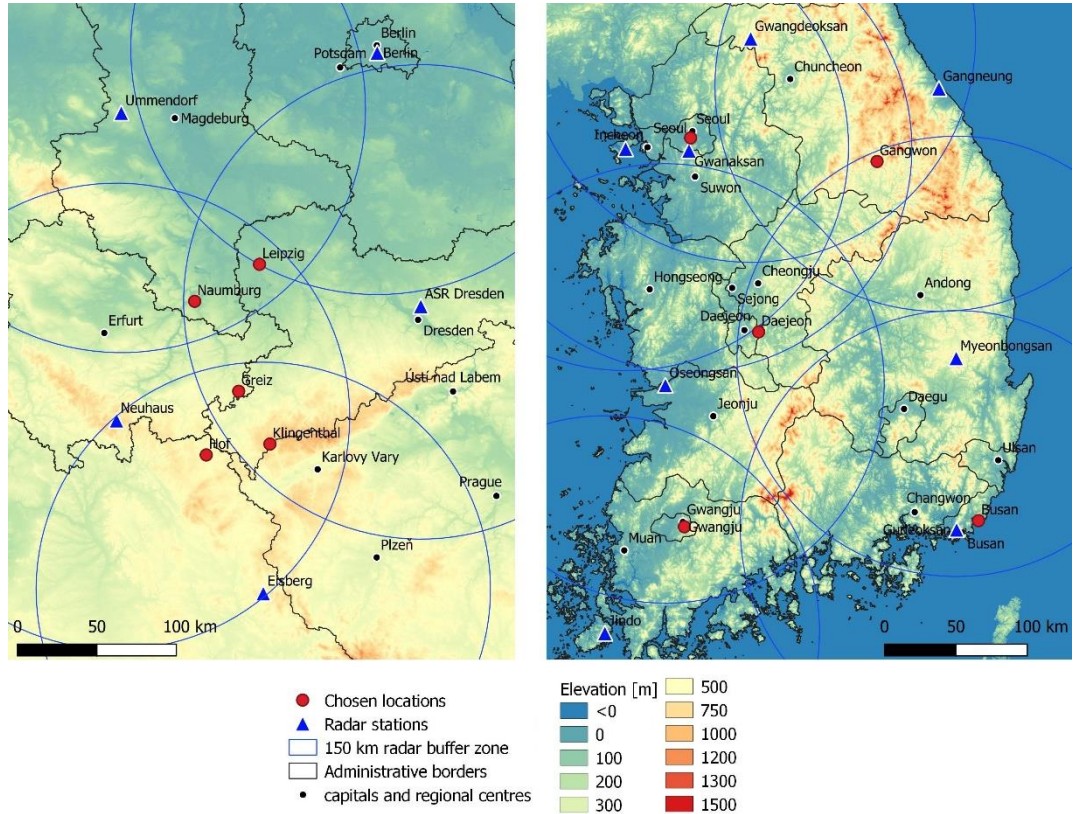

**Fig 5. Overview map of the chosen locations and corresponding radar stations for Germany (left) and South Korea (right).**

Radar data was utilised as a high-resolution reference dataset. For the German sites, Radar-based Precipitation Climatology Version 2017.002 dataset (available at https://opendata.dwd.de/climate_environment/CDC/help/landing_pages/doi_landingpage_RADKLIM_RW_V2017.002-en.html) was used (Winterrath et al., 2018). The dataset is as composite available on a 1x1 km grid for the time period of



2001-2020 and temporal resolution of 5 min. It represents a product of the RADOLAN method, where the precipitation sums from the radar-based precipitation estimates are adjusted using measurements from conventional gauges (Winterrath et al., 2017). Thus, for the chosen locations, data from five overlapping radar stations are merged (Berlin, Dresden, Eisberg, Neuhaus and Ummendorf). For the South Korea sites, the composite radar rainfall product CM1 (available at

https://data.kma.go.kr/data/rmt/rmtList.do?code=11&pgmNo=62) from the Korea Meteorological Administration (KMA) was used. The dataset merges observations from 11 radar stations and has a 1x1 km grid with temporal resolution of 10 minutes covering the period from 2009 to 2019. As a quality control, the dataset passed through the algorithm of the Open Radar Product Generator. This algorithm uses corrected reflectivity, which filters ground echoes and detects non-precipitation echoes thereafter removing them based on the criterion related to the difference of reflectivity at the upper and

lower side from a certain altitude (Park et al., 2014).

For the German sites CanESM2 projections for RCP2.6 and 8.5 scenarios downscaled with the EPISODES model were used for the period 2020 to 2100. The Canadian Earth System Model (second generation) with a T63 (~1.9°) resolution consists of the physically coupled atmosphere-ocean model CanCM4 coupled to terrestrial carbon and ocean carbon models (Arora et al., 2011). EPISODES is an empirical-statistical downscaling method (Kreienkamp et al., 2019). It implements a two-step

procedure. The first part provides day-by-day meteorological information on regional scales, which are used as an input to the second part producing synthetic time series via a weather generator. The target grid corresponds to the EURO-CORDEX resolution of 0.11° (~12 km). Original climate projection data with daily resolution were bias-corrected using monthly quantile-mapping on the 1x1 km interpolated station-based RaKliDa dataset (Kronenberg and Bernhofer, 2015).

For the Korean sites, high-resolution data from the Korea Meteorological Administration with SSP1-2.6 and SSP5-8.5

scenarios covering a period from 2020 to 2100 was chosen (available at http://www.climate.go.kr/home/CCS/contents_2021/Kma_climate_RCP.html). The dataset is based on the UKESM1 global and a set of dynamic and statistical downscaling models. UKESM1 from Met Office Hadley Center is based on the HadGEM3-GC3.1 model (atmosphere-land-ocean-sea ice) combined with the JULES terrestrial and the MEDUSA ocean biogeochemical models (Sellar et al., 2019). This global model has 135 km resolution which is dynamically downscaled to

25 km through ensemble mean (Kim et al., 2022) of the 5 regional climate models (HadGEM3-RA, RegCM4, SNURCM, GRIMs, and WRF) participating in the CORDEX-EA II project. Finally, the regional climate model was downscaled to 1 km grid by the PRIDE model (Kim et al., 2016b) using ground observation data and the Barnes approach (Barnes, 1964).

For all sites the nearest grid cell to the study sites were taken from the respective datasets. For the sake of consistency between German and Korean radar datasets, the German one was aggregated to 10 min resolution. Comparability between

different climate projection generations (CMIP5 and CMIP6) and thus between respective RCP and SSP concepts is also preserved (O'Neill et al., 2016): RCP-2.6 and SSP1-2.6, RCP-8.5 and SSP5-8.5.



### Model validation and evaluation of climate projections

Validation of both methods was done by comparison of disaggregated and original radar datasets, where the first was produced using original radar values aggregated to a daily scale. Although both methods are of stochastic nature, the straightforward comparison of the disaggregated time-series between both is not possible, as LetItRain acts as Monte-Carlo precipitation simulator, while WayDown keeps daily precipitation sums consistent with the input data, however implying a stochastic component. Thus, the following statistics were chosen to compare models on monthly and annual scale for the whole and non-zero time-series: mean, variance, transition probabilities of Markov Chain, autocorrelation function values, proportion of wet period, frequency and quantiles of extreme events of various magnitude and annual maxima. Furthermore, for each station the time-series of 1000-year length equivalent were generated to test the possible variability of model simulation statistics.

Disaggregated time-series of the future climate projections were evaluated with regard to change factors and annual extremes. Change factors were calculated as a simple ratio between future and historical period for the following precipitation statistics: mean, variance, autocorrelation function, proportion of wet period and 99% extreme quantile.

### Results and discussion

#### Validation of the methods with radar data

##### Visual inspection of observed and disaggregated model events

A direct comparison of observed radar and disaggregated precipitation events using standard methods like e.g. time-series overlap plots or correlation coefficient is not reasonable due to the stochastic nature of the models. However, a qualitative visual mapping is possible. For that, 5 random daily events from typical winter (February) and summer (July) months were selected, with daily sums close to respective daily mean values (Fig. 6 and Fig. 7, left panels). Corresponding disaggregated events were randomly picked from the models' ensembles considering same (for LetItRain - close) daily sums and similar daily maxima with previously chosen radar events (Fig. 6 and Fig. 7, middle and right panels).

For Leipzig, a typical winter event has a sum of 2.8 mm and a maximum intensity of 0.2-0.6 mm/10-min (Fig. 6). In the summer time higher values are normally observed - 5.2 mm daily mean and maxima of 0.5-2 mm/10-min (Fig.7). For Seoul, however, the difference between February and July is more prominent and events are generally more autocorrelated. Typical means are 4.1 and 18.9 mm for winter and summer months, respectively, and typical maxima range from 0.2-0.7 mm/10-min in February to 1-8 mm/10-min in July.

From visual inspection both models show satisfactory results in replicating 10 min radar for both cities capturing typical magnitudes, variability and alignment of sub-daily rain events, especially considering the variability in precipitation regime, seasonality and radar precision.





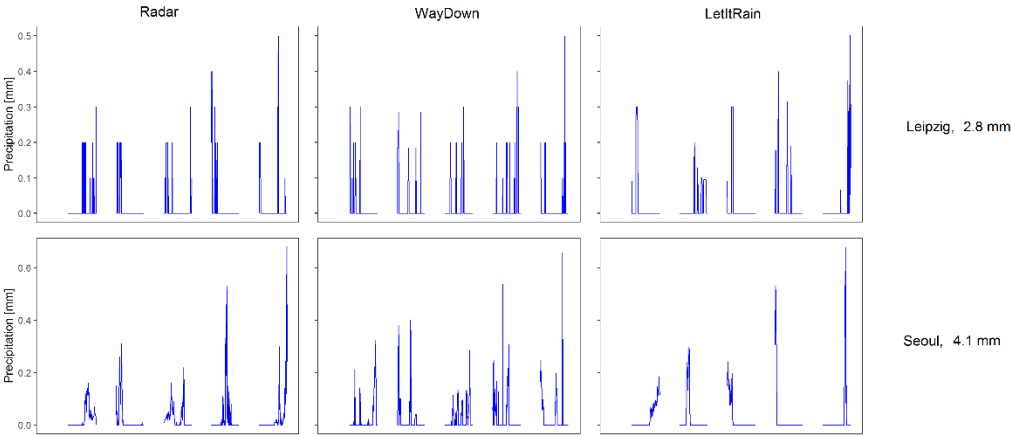

**Fig 6. Five winter (February) sample events with typical daily sums for Leipzig, Germany and Seoul, South Korea.**

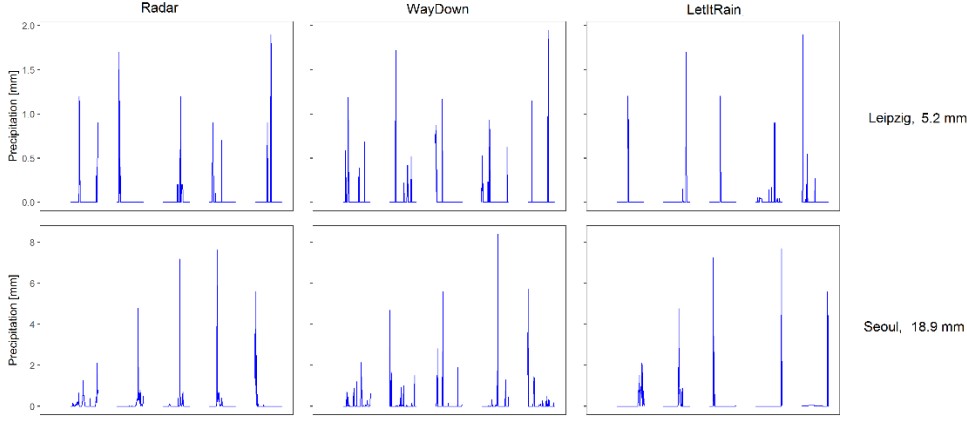

**Fig 7. Five summer (July) sample events with typical daily sums for Leipzig, Germany and Seoul, South Korea.**

### Comparison of main statistics

Validation results are presented with monthly statistics plots for Leipzig and Seoul (Figure 8, for other stations see Appendix A1) and annual statistics for all stations (Table 1). According to radar data, mean and variance for Korean stations were found to be 2-10 times higher than for German stations (0.011 and 0.039 mm for mean and 0.010 and 0.074 mm2 for

variance, respectively). Transition probabilities from 'dry' to 'wet' conditions are similar for both countries (0.01-0.05), while 'wet-wet' persistence probabilities for Korean locations (0.75-0.90) are slightly higher than for German stations (0.60-0.70). Autocorrelation function values were also found to be higher for Korean sites. For example, for 10 and 60 min lag, monthly variance between autocorrelation values was found to be 0.5-0.75 and 0.15-0.35, respectively, for Germany and 0.60-0.80 and 0.20-0.50, respectively, for South Korea. Typical proportions of wet periods for Korean stations lie between

13% and 19%, while German stations showed much lower values (3-4 %) which is in line with the differences of frontal precipitation behaviour for two countries.





Statistical moments of the first and second order for the full time-series length were well represented by both LetItRain and WayDown. This includes not only the values of annual mean and variance, but also replicating the pronounced seasonal cycle. Only minor deviations for both models were observed mostly for the summer months. A perfect match of means for
radar and WayDown can be explained by the nature of the method as it keeps the daily sums consistent. However, for the non-zero time series, the difference between models' behaviour is noticeable. WayDown overestimated non-zero mean for both countries by 0.05-0.4 mm especially in summer months. It is consistent with the simultaneous underestimation of dry-wet period proportions while keeping the daily sums preserved and can be explained by two reasons. The first reason is the systematic overestimations of  sampling from the fitted 2D empirical copulas, which apparently do not possess a good
representation of the real precipitation behaviour. Secondly, the assumed 10% convergence to the daily sum of the disaggregated values used for the final adjustment procedure could be probably too high and needs to be reduced for more precise estimations. This will lead however to considerable increase of computation time. LetItRain on the other hand, generally underestimated non-zero mean by 0.05-0.1 mm for both countries. This can be explained by the model fitting process, which is trained to replicate the mean of observed rainfall and does not consider non-zero mean statistics. Non-zero
variance on the other hand was better represented by both methods than non-zero means however with the same general behaviour for both methods.

Monthly variations of Markov Chain transition probabilities from dry to wet and persistence of wet states were better modelled by WayDown, which was expected, since transition matrices are directly incorporated in the method for the binary time-series sampling. However, systematic underestimations of 5-10% for dry-wet and wet-wet states probabilities were
found for all stations. This is probably due to the shortcomings and assumptions of the radar sub-daily events subset procedure (which is used to fit the Markov Chain) in WayDown based on a certain daily precipitation sum and month. Namely, the number and representativity of this subset is directly limited by the radar time-series length and thus has a significant influence on the accuracy of transition matrix estimations. LetItRain performed multidirectional and in general did not replicate this statistic well including seasonality. It showed significant underestimations of dry-wet transitions and
overestimations of persistence probabilities for almost all stations. Especially huge mismatch was found for persistence probabilities. Although dry-dry and wet-wet transition probabilities are included in the Poisson cluster rainfall model (Cowpertwait et al., 1996), they are estimated with analytical equations based on the proportion of dry periods at several aggregation intervals. In this study, however, this parameter was not calibrated.

Autocorrelation function values of different lags were simulated by both methods with different qualities for both countries.
WayDown showed a good match for 10 min lag for non-zero time series, while with larger lags the correlations were underestimated, which is especially noticeable for Korean sites. This is due to incorporation of only 2D rather than higher-dimensional copulas for precipitation sampling which indirectly affects autocorrelation of higher orders. Systematic underestimation on autocorrelation for the full time series (0.1-0.3) in WayDown probably originates from the underestimation of zero precipitation intervals (proportion of wet period). LetItRain, on the other hand, depicted better for





the full time-series and higher lags, while overestimating (for German stations) or missing the annual cycle (for Korean stations) the lag-1 autocorrelation values. As LetItRain directly incorporates autocorrelation parameters of several lags in the model setup, it explains the better model match. The model, however, could not be fitted perfectly for that statistics since the greater weight is set to mean and variance rather than autocorrelation (see Supplementary). Non-zero autocorrelation followed the full time series values with slightly worse performance for both models.

Proportion of wet periods for South Korea was underestimated by WayDown (approximately by 5%) while LetItRain showed a good agreement. For the German stations, WayDown showed minor underestimations (< 1%), while LetItRain, on the other hand, generally overestimates the proportion by 1-3%. For WayDown, the systematic errors are connected to the problem of binary time-series sampling discussed above and directly explained by minor underestimations of dry-wet transition probabilities, which were also slightly higher for the Korean sites. In the case of LetItRain, the minor deviation

occurred because the model was not calibrated to perfectly reproduce the proportion of wet periods.

Overall, based on the variety of analysed statistical characteristics, it could be concluded that LetItRain delivers much higher variability for all variables compared to WayDown, especially for Korean stations. Since LetItRain represents a stochastic rain generator, rather than a pure disaggregation model as WayDown, it naturally shows a wider ensemble variability. Furthermore, as the model requires fitting of multiple statistical parameters, it was found that the calibration procedure

struggled to find optimal parameters for a few stations and months (i.e. see non-zero mean value and proportion of wet periods for June in Leipzig), probably due to the high precipitation variability and shortage of reference radar time-series.

**Table 1. Summary of annual statistics for disaggregated and original radar data.**

| | | Greiz | Hof | Klingenthal | Leipzig | Naumburg | Busan | Daejeon | Gangwon | Gwangju | Seoul |
|---|---|---|---|---|---|---|---|---|---|---|---|
| Mean [mm] | Radar | 0.010 | 0.010 | 0.011 | 0.012 | 0.011 | 0.038 | 0.040 | 0.026 | 0.048 | 0.043 |
| | WayDown | 0.010 | 0.010 | 0.011 | 0.012 | 0.011 | 0.038 | 0.040 | 0.026 | 0.048 | 0.043 |
| | LetItRain | 0.010 | 0.009 | 0.010 | 0.012 | 0.010 | 0.037 | 0.042 | 0.031 | 0.049 | 0.040 |
| Variance [mm2] | Radar | 0.009 | 0.008 | 0.010 | 0.012 | 0.009 | 0.069 | 0.074 | 0.036 | 0.092 | 0.099 |
| | WayDown | 0.009 | 0.008 | 0.009 | 0.012 | 0.009 | 0.073 | 0.066 | 0.035 | 0.093 | 0.092 |
| | LetItRain | 0.008 | 0.007 | 0.010 | 0.010 | 0.008 | 0.062 | 0.075 | 0.042 | 0.089 | 0.085 |
| ACF lag 10 min [-] | Radar | 0.66 | 0.66 | 0.62 | 0.66 | 0.64 | 0.72 | 0.73 | 0.74 | 0.68 | 0.73 |
| | WayDown | 0.56 | 0.58 | 0.55 | 0.57 | 0.55 | 0.73 | 0.73 | 0.72 | 0.71 | 0.73 |
| | LetItRain | 0.75 | 0.75 | 0.75 | 0.79 | 0.67 | 0.77 | 0.86 | 0.80 | 0.77 | 0.81 |
| Wet period proportion [-] | Radar | 0.034 | 0.031 | 0.035 | 0.038 | 0.034 | 0.14 | 0.19 | 0.13 | 0.19 | 0.19 |
| | WayDown | 0.027 | 0.025 | 0.028 | 0.030 | 0.028 | 0.095 | 0.128 | 0.079 | 0.12 | 0.13 |
| | LetItRain | 0.041 | 0.036 | 0.038 | 0.047 | 0.043 | 0.14 | 0.275 | 0.171 | 0.24 | 0.23 |



**Fig 8. Comparison of monthly statistics between disaggregated and original radar data for Leipzig and Seoul.**

### Representation of extremes

Along with the replication of monthly and annual statistics, it is also important for downscaling models to keep consistency of extreme precipitation frequencies and magnitudes.

Absolute frequencies of 10-min precipitation extremes for reference radar and disaggregated datasets in Leipzig and Seoul, normalised to the number of overshooting events per 100 years, are presented in Figure 9 (left panels). Due to the differences in climate, extreme frequencies in the intensity interval from 5 to 15 mm in Seoul are 5-50 times higher than for Leipzig. For example, the estimated frequency of 10 mm / 10 min events for German sites is in the range of 15-35 per 100 years, while





for South Korea sites it yields 155-620 (Appendix A2). For all the German sites, both models behaved in a similar way. The ensemble median of frequencies until 2-3 mm / 10 min is overestimated up to 20%, while for higher thresholds frequencies

are underestimated, with increased magnitude towards higher values. Noteworthy, although LetItRain showed slightly higher underestimation of the frequencies for the median, its ensemble bandwidth covered zero relative difference with radar data for all German stations and intervals (Figure 9, right panels, Appendix A2). For the Korean stations, WayDown showed a similar over-underestimation of extremes around the threshold of 2-5 mm / 10 min. LetItRain behaved for each Korean site differently, however generally depicting lower deviations from radar data than WayDown.

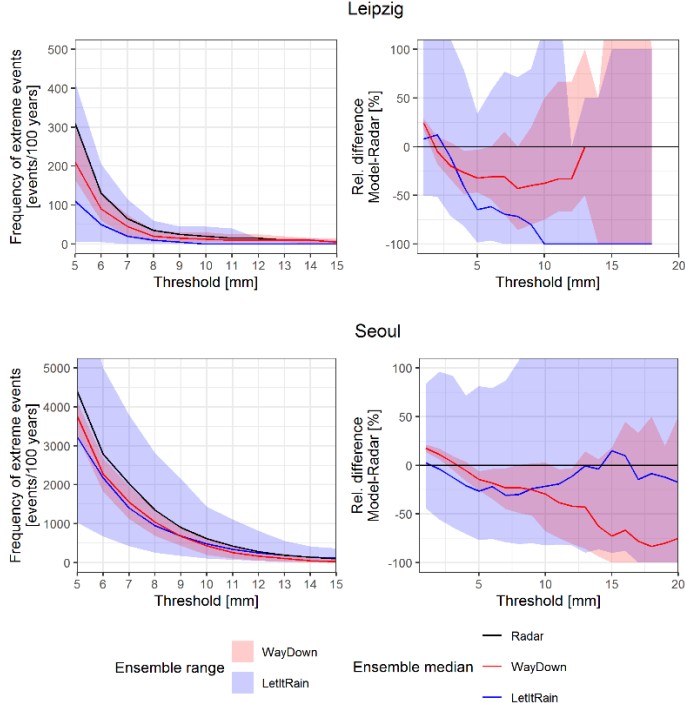


**Fig 9. Absolute and relative difference in frequency of extreme precipitation for original and disaggregated radar data for Leipzig and Seoul.**

Extreme precipitation quantiles (0.99 to 0.99999) for reference radar and disaggregated datasets for Leipzig and Seoul are presented on Figure 10 (left panels) and for the other stations in Appendix A3. For example, estimated empirical 10-min

quantiles with 0.99 probability for German and Korean sites lie in approximately the same range of 1.8-2 and 2.1-3.1 mm, respectively (Appendix A3). Results look plausible with regard to the difference in climate and time-series lengths. For the German stations, WayDown overestimated extreme quantiles until 0.99-0.999 probabilities, thereafter producing slight underestimations with deviations of approximately 20% from the observed data. LetItRain, on the other hand, showed good fit on extreme quantiles up to the 0.99 percentile and underestimations for the higher ones. For the stations in South Korea,

WayDown possesses overestimation of extremes up to the 0.999 percentile, underestimating them further up to 40%.





LetItRain showed slight underestimation of quantiles except for Busan. Similarly to extreme frequencies, LetItRain showed much wider ensemble bandwidth with errors up to hundred percent for rare events (Figure 10, right panels).

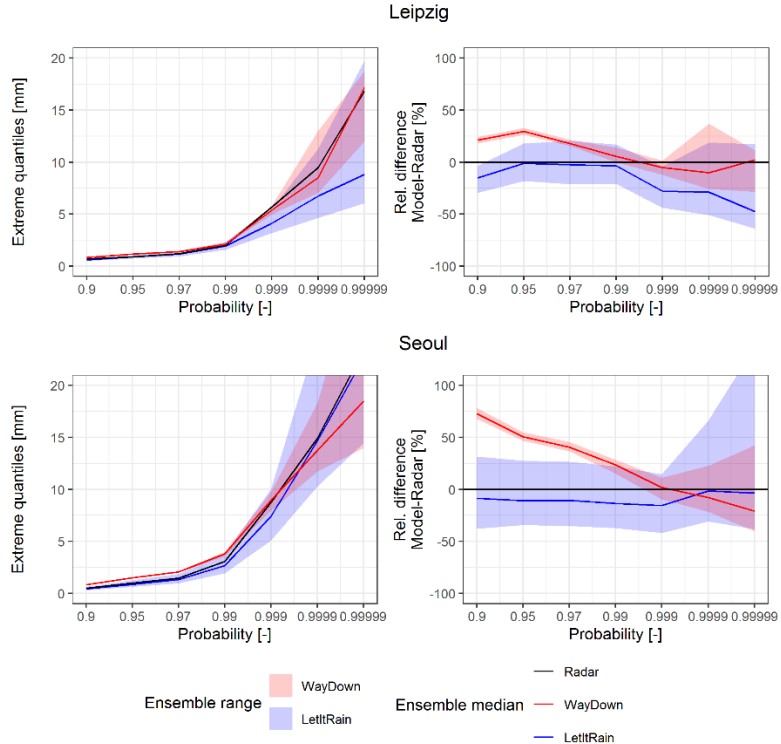

**Fig 10. Absolute and relative difference in extreme precipitation quantiles for original and disaggregated radar data for Leipzig and Seoul.**

Variability of annual maximum precipitation for reference radar and disaggregated datasets for all stations is shown in Figure 11. According to radar data, median annual maximum values for German sites are 6.3-7.5 mm / 10 min, while for the Korean locations typical maximum values are almost twice higher (10.9-15.0 mm / 10 min). For all German stations, plots show for both models systematic underestimation of annual maxima. The distributions for all 1000-year-equivalent time-series depict higher positive skewness and median values of 4.9-5.9 mm / 10min for WayDown and 4.3-5.5 mm / 10 min for LetItRain were found. On the other hand, for the stations in South Korea, no systematic underestimation was noticed. Simulations from LetItRain were closer to radar data with median values of 10.0-16.8 mm / 10 min, while WayDown showed deficient agreement (median 7.7-12.8 mm / 10 min).





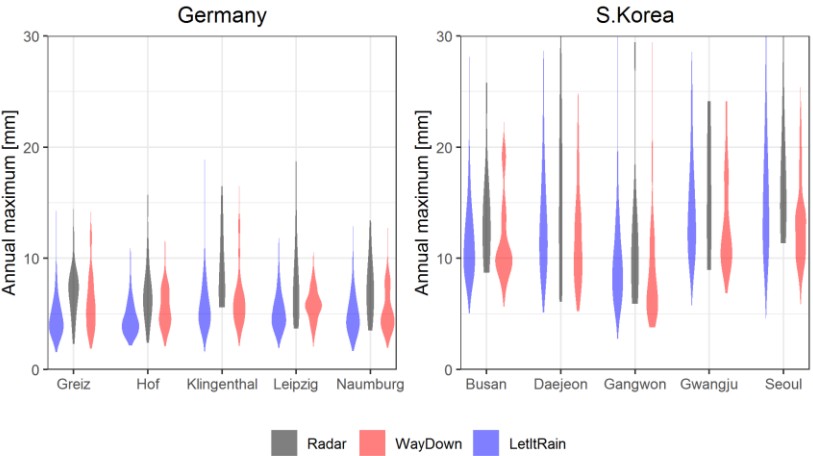

**Fig 11. Violin plots with annual 10-minute precipitation maxima for original and disaggregated radar data for Germany and South Korea.**

Problems of the WayDown approach shown above regarding representation of the extremes (over and underestimation of frequencies and quantiles around certain thresholds and underestimation of annual maxima) originate from the precipitation sampler which is based on 2D empirical copula. While serving as a simple and not site-specific universal method, which does not require calibration, it showed satisfactory and robust results regarding main statistics but naturally reveals a number of shortcomings one of which is inaccuracy in extreme precipitation representation. The issue (in addition to the ones mentioned in the previous section) can be solved by the application of an improved copula sampling. Nesting copulas might improve autocorrelation, while application of parametric copulas will provide a better fit for extremes. This however will lead not only to an increase of computation time, but also to the challenges of a better copula family choice and fitting procedures, which are so far tricky to implement in an automatic way without user intervention and control.

The mismatch for the extremes in the LetItRain results can be explained by the shuffling algorithm of the model. The model first simulates rainfall using a Poisson cluster-based rainfall model, and then uses an algorithm to rearrange the rainstorms (see Supplementary (Vorobevskii, 2023)). It takes into account the correlation between the rainstorms in the observed rainfall and accordingly calibrates the 'deg' parameter. It means that storms are more likely to be rearranged in that storms with greater rainfall amount flock together in a 1000-year simulation. As we segmented the generated time-series into several ensembles, the storms with large extreme values are likely to belong to only a few ensemble members, therefore leading to an imbalanced distribution of extremes between the ensembles.

### Change factors between radar and climate projections

Change factors for the main statistics between climate projections and radar data for two scales are presented in Figure 12. Change factors for the German stations on a daily scale showed mostly positive trends for both RCP scenarios, except for



Leipzig and Naumburg, where trends were multidirectional. Korean stations, on the other hand, were mainly characterised by negative trends, except for autocorrelation, where change factors were found to be positive for all sites.

It might be expected that the change factors from disaggregated time-series of climate projections will follow a similar trend as for a daily scale. They are based on basically the same input data used for calculations of daily factors (upscaled radar and disaggregated climate projections). Moreover, the LetItRain method directly incorporates daily-scaled change factors for the application to climate projections (although only for the mean). It was found that the difference in change factors between scales is much higher, than between the two RCP scenarios (for the same model). Furthermore, the agreement between both 10-min datasets is higher than for the daily scale data. Nevertheless, the direction of trend for the daily scale was in a better agreement with 10 min data for Korean, compared to German stations simulated with both models, especially when analysing mean, variance and proportion of the wet periods. Only three German sites showed similarities between scales and only for few statistics (autocorrelation and proportion of the wet periods).

For all German stations, both models resulted in a non-existing or negative trend for mean and variance (both full and non-zero time-series), while showing mostly positive trends for autocorrelation and proportion of wet periods. Further, extreme precipitation of 99% was simulated differently by the methods. For Korean stations models ended up with negative trends for mean and variance and proportion of the wet period, while change factors for non-zero mean and variance, as well as extreme precipitation increased for the future period. Trends for the autocorrelation did not agree between the methods. Finally, it was found that except for a few cases, WayDown delivered lower change factors than LetItRain.

Cross-analysis of the disagreement between two scales of change factors and problems of both methods revealed and discussed in the validation section does not show clear patterns either in the context of countries or statistics. Thus, the identified trend differences between daily and 10-min scales could not be solely explained by the shortcomings of the methods. Furthermore, as for most of the sites and statistics both models showed similar behaviour of change factors, both can express the possible reality of the future precipitation changes for the finer scale.







**Fig 12. Change factors for the main statistics between climate projections and radar data on daily and 10 min scale for Germany (left) and South Korea (right).**



### Extremes in climate projections

We compared the behaviour of annual maxima for all stations on daily (Fig. 13, upper panel) and 10-min scale (Fig. 13, lower panel) to check indirectly the plausibility of the extremes in the generated time series for the climate projections. For that, the whole length of available time-series was used (80 years for climate projection data and 20 / 11 years for radar data for German and Korean stations respectively) As the in-depth discussion on the trustworthiness and quality of input precipitation climate model data is not the topic of the study, we just state here that on the daily scale radar showed significantly higher maxima than both climate model outputs for all Korean and two German stations (Leipzig and Naumburg). The differences between the two scenarios are in general minor: seven out of ten stations deliver a slightly higher median of annual maxima for RCP 8.5 scenario and for the rest three the values are similar. Variability of maxima, expressed in the distance between main quartiles (25-50%), for the radar data is up to 3 times (1.5 for German sites) higher than for the climate projections. Between two scenarios, RCP 8.5 possesses higher inter-quartile variability for the majority of stations.

For the 10 min scale, median annual maxima from radar datasets exceeded the ones from climate projections for all stations except Gwangju and Seoul, where medians from RCP 8.5 scenarios have similar values. It is mainly driven by underestimation of extremes in the original daily climate projection data. Except for a few cases, both LetItRain and WayDown showed similar behaviour regarding relative differences between radar data and RCP scenarios of disaggregated climate projections (median, variance) compared to daily data. Comparison between RCP scenarios for the same disaggregation model did not show systematic patterns, e.g. that RCP 8.5 will deliver higher annual maxima, which agrees with the results for a daily scale. Comparing the two models, LetItRain delivered higher variability and noticeably absolute values for six stations (four Korean and two German), which could not be referred to as a systematic disagreement between two methods. This difference also needs to be backed up by the fact that LetItRain acts as a rain generator model and already showed much higher variability between model realisations in the validation part. Thus, it could demonstrate the same feature for 80-year long climate projection time-series and taking an ensemble rather than one realisation can narrow the inconsistency in annual maxima representation between two methods.





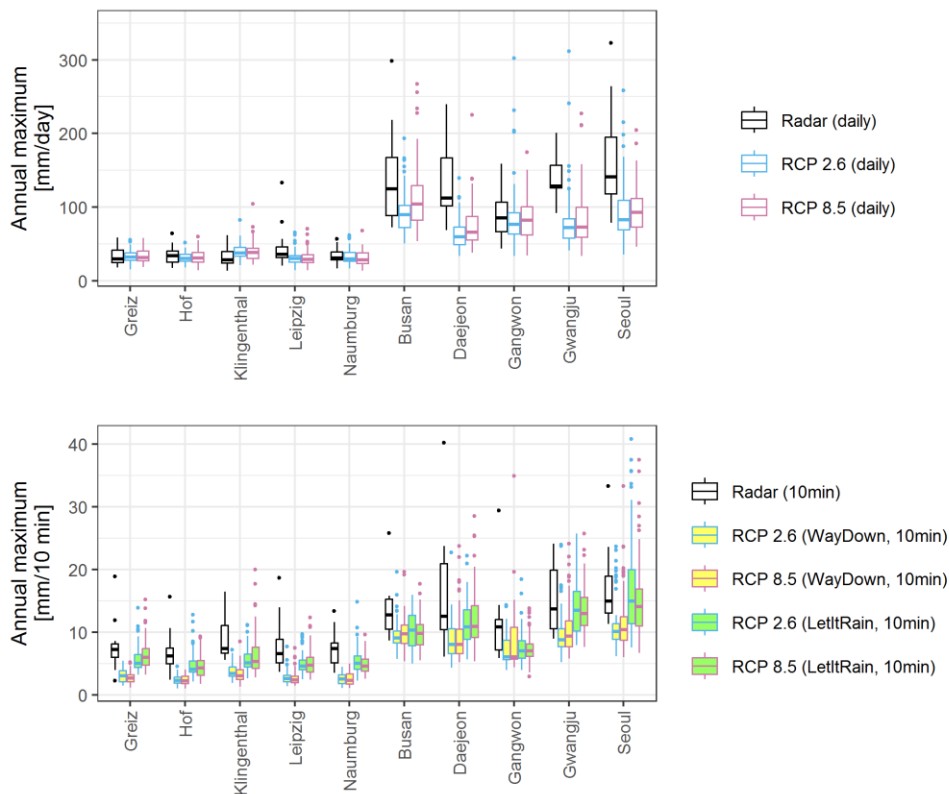

**Fig 13. Annual maxima of climate projections and radar data on daily and 10 min scale for Germany and South Korea.**

**Conclusions and outlook**

In this study, we presented and discussed two different methods to disaggregate the daily output of projected precipitation
data to sub-hourly scale. Although both of them belong to a stochastic class of models, the first is a pure disaggregation
model which keeps daily sums consistent (WayDown), while the second represents a stochastic rain generator, which mainly
focuses on the replication of time-series statistics (LetItRain).

We validated both models using radar data from 10 stations located in Germany and South Korea. It should be mentioned
however, that both models are not limited to this resolution and can reproduce statistics up to 1 min, if respective reference
data is provided. The success of the validation was evaluated by the matching of multiple statistics calculated from the
475 original 10 min radar data and disaggregated time-series, where the same upscaled daily radar data was used as model input.
To account for possible model uncertainty, the disaggregation procedure was replicated several times in order to get a 1000-
year equivalent ensemble output. With regard to ensemble median, WayDown showed better results for monthly mean and
variance (including non-zero time series), transition probabilities of the Markov Chain, 1-lag autocorrelation and proportion
of wet period, sometimes over- or underestimating the absolute values, but following the sub-annual cycle. LetItRain better





480 replicated autocorrelation values of higher lags (up to 60 min) and depicted good results for mean and variance. Although some other characteristics on the annual scale proved to be properly simulated, the model struggled to fit the seasonal course for many statistics. Furthermore, ensemble variations of LetItRain were found several times higher than for WayDown. Both methods showed better results for German rather than for Korean stations. The frequency of extremes was generally underestimated by the models for thresholds of 2-3 mm and 2-5 mm for German and Korean sites, respectively. For the

485 German stations, WayDown overestimated extreme quantiles up to 0.99-0.999 probabilities, afterwards it showed slight underestimations, while LetItRain demonstrated underestimation of all percentiles. For the stations in South Korea, both methods overestimated extremes up to the 0.999 percentile, thereafter underestimating them.

 Further, we applied the models to climate projection data and compared change factors and extremes to radar data between two time scales. For the majority of the cases, change factors for daily and 10 min resolutions do not follow each other. In

490 fact, they depicted similar values for the same model and RCP scenario. Consistent positive and negative trends confirmed jointly by models and daily data were found for three stations in Germany (for autocorrelation and proportion of wet periods) and five stations in South Korea (for mean, variance and proportion of wet periods), respectively. Both models showed similar quantile values of annual maxima for the majority of the stations for both RCP scenarios in comparison to daily scale. A systematic underestimation of annual 10 min maxima was found for both methods compared to radar data. Mainly,

495 this is due to their underestimation in the original daily climate projection data. Finally, the application of the disaggregation models on the climate projection data should be done with caution, especially for the case when statistics of the current period are preserved and assumed for the future. Although some indications of the possible changes in sub-daily statistics exist (Meredith et al., 2019), there is still not enough consistent knowledge to prove it (e.g. high time resolution climate projection simulations).

500 The comparison of two methods clearly revealed both similarities and differences that can provide crucial information in the choice of disaggregation model type for producing fine-scale future rainfall, which few studies have yet addressed. Moreover, as it was shown and discussed in the validation and application part, that the presented models have potential for improvements which most likely will result in a higher quality of the disaggregation. For WayDown this includes incorporation of parametric nested copulas in the precipitation sampler and accounting for change factors between radar data

505 and climate projections on a daily scale to apply these trends in the disaggregation process. For LetItRain the statistics additional statistics can be included in the calibration procedure (e.g. transition probability). Further, the effect of skewness at a fine time scale in the calibration procedure should be investigated more deeply for accurate replication of extreme rainfall. Moreover, future high-resolution climate model projections will be able to generate data, allowing deducing and incorporating higher-order moment statistics into the presented downscaling models. Finally, integration of the methods,

510 which account for spatial correlation and thus allowing shifting from point to spatial scale, will be beneficial as well. Lastly, in order to prove the utility of simulation approaches, a downstream application of simulated data in urban hydrological models would be beneficial, too.



**Data and Code availability**

Authors fully support open-source and reproducible research. WayDown is available on GitHub
https://github.com/hydrovorobey/WayDown repository (CC BY-NC-ND 4.0). LetItRain is available on the Hongik
University's Hydrology Innovation Laboratory web page (https://sites.google.com/site/hihydrology/projects). Locations of
the stations, input datasets, simulation results, and R-scripts to reproduce figures and tables for the manuscript are available
under the following HydroShare composite resource https://doi.org/10.4211/hs.9322e1ef25e04822a759c515795642e1
(Vorobevskii, 2023).

**Author contribution**

Conceptualization VI, KD, KR; data curation VI and PJ, formal analysis VI, methodology VI and PJ; visualisation VI;
writing: original draft preparation VI, PJ, KD, BK, review KR. VI and PJ have the equal contribution to the paper.

**Competing interests**

The authors declare that they have no conflict of interest.

**Funding**

This research was supported by the DAAD with funds from the German Federal Foreign Office; This research was supported
by the Basic Research Laboratory Program (Grant Number: 2022R1A4A3032838) through the National Research
Foundation of Korea (NRF) funded by the Ministry of Science and ICT; This work was supported under the framework of
international cooperation program managed by the National Research Foundation of Korea (NRF-2021K2A9A2A15000179,
FY2021).





## Appendix

### A1. Comparison of monthly statistics between disaggregated and original radar data (remaining stations)







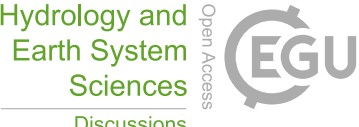

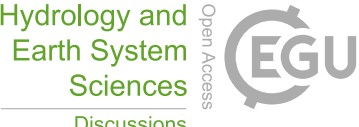





**A2. Absolute and relative difference in frequency of extreme precipitation for original and disaggregated radar data (remaining stations)**



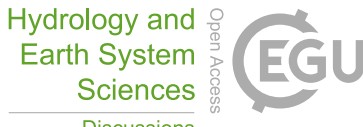

## A3. Absolute and relative difference in extreme precipitation quantiles for original and disaggregated radar data (remaining stations)

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
