# Peer review of "Simulating sub-hourly rainfall data for current and future periods using two statistical disaggregation models - case studies from Germany and South Korea"

_Hydrology and Earth System Sciences, 2023_

## Author Comment (AC1)

Review 1

| I do not understand how you obtain the RCP2.6 and RCP8.5 radar data. Is this what you refer to in lines 242–243 and lines 251–252 as the "1x1 km interpolated station-based RaKlida dataset" and "the regional climate model was downscaled to 1 km grid by the PRIDE model"? If so, stating that it is RCP2.6 and RCP8.5 radar data is misleading, because it is really just downscaled climate model data and not the observed radar data. | We guess a misunderstanding occurred here. We use both radar observations 5/10 min data (lines 222-235) and daily climate projection (CP) modelled data with 2 RCPs scenarios (lines 236-252). Radar is used for models' validation in the first part of the results, while CP are used to show the models' application in the second part. Additionally, radar data is used to derive precipitation characteristics for the model (like a training dataset), which then are used for CP disaggregation. We did not find in text where we mix terms of CP and radar. Maybe the Reviewer refers to the legend of Fig.12, where namings like e.g. 'RCP 2.6 - Radar (daily)' occurs. Here '-' sign means 'between', referring to the change factor between CP and radar data on a daily scale. We suggest changing it to 'and' to avoid confusion (e.g. 'Daily scale: RCP 2.6 and Radar'). |
|---|---|
| I assume you are using the same downscaled data to ingest into WayDown and LetItRain as RCP2.6 and RCP8.5 radar data, correct? If so, please state that explicitly in the methods. | Agreed, will be clarified. |
| Since you are using different models and different downscaling methods for climate projections and Germany and South Korea, I do not think it is fair to compare how well the disaggregation models compare against each other in those two locations because they have different model data they are ingesting. However, when you compare the "change factors" *within* each location, that is OK. | Agreed, the result section containing CP disaggregation comparison will be corrected. |
| Please improve the resolution of the figures, notably Fig.5's legend, Fig. 6 and 7. | Agreed, all produced figures have originally 300 dpi resolution, we guess the resolution was muffled due to word-to-pdf conversion. Will be double check it during the article production process. |
| Line 144: When you say "binary 5-minute precipitation", I am guessing you are referring to whether it rained or not? If so, please make that clear. | Agreed, will be clarified. |

| | |
|---|---|
| Figure 2: Please make this figure caption clearer. I recommend referring to the appropriate panel after the corresponding text. I am guessing "daily value of 15 mm" refers to the left panel? | Agreed, will be clarified. 15 mm refers to daily value, first panel represents uniform distribution of these 15 mm for 10 min intervals, second - binary event sampling and third how the model 'fills' binary data with real values and corrects them. |
| Line 202: Do you calibrate the model using the raw future climate data? Please state what you use for calibration. | We did not use the raw future climate data for model calibration. To calibrate the model for future climate, we follow the procedure as follows. At first we derived linear regression between rainfall statistics using high-resolution reference data (radar). Secondly, the climate change signal from the climate data is reflected using a change factor. Change factor is defined as the ratio of mean between historical and future periods and is used to adjust future 10-minute precipitation mean. Thirdly, future rainfall statistics are estimated using the obtained linear regressions from radar data and calculated change factor. The future 10-minute rainfall mean is estimated by multiplying the observed 10-minute rainfall mean by the change factor. Afterwards, other future rainfall statistics are estimated using this future 10-minute rainfall and linear regression between statistics. The set of estimated future rainfall statistics is used for calibrating the model. This procedure is described in manuscript L183-L201. |
| Line 232–235: I am glad you use an algorithm to correct reflectivity, but I am wondering if you considered the impact of beam-blockage due to the mountains in South Korea? Did you account for this? | Yes, the radar dataset accounts for this effect. We elaborated on the dataset description (Radar Quality Control) in more detail. Firstly, in this algorithm, corrected reflectivity data is utilised. This corrected reflectivity data is obtained by applying the Gaussian Model Adaptive Processing (GMAP) filter (Siggia and Passarelli, 2004), which corrects for echoes caused by the surrounding terrain, such as beam blockage due to mountainous terrain, in the reflectivity data. Then, this algorithm detects non-precipitation echoes thereafter removing them based on the criterion related to the difference of reflectivity at the upper and lower side from a certain altitude (Park et al., 2014). Siggia, A. D., & Passarelli, R. E. (2004, September). Gaussian model adaptive processing (GMAP) for improved ground clutter cancellation and moment calculation. In Proc. |

| | ERAD (Vol. 2, pp. 421-424). |
|---|---|
| Figure 6: I recommend labeling the five separate events with text or plotting the five different events in five different colors. Also, how do you determine which events count as separate, as some have multiple peaks in precipitation? | Agreed, the figure will be improved to have more clear separation between picked days. Since this figure shows just an example of daily disaggregation, sub-daily events here are not separated. To avoid confusion we suggest to change caption name (events -> days with precipitation) |
| Lines 295–300: Please refer to the subplots in the above figure to make the text easier to interpret. | Agreed, will be added. |
| Line 389–390: I recommend putting more detail in the methods to explain how you obtained the 1000-year time series. | Agreed, will be elaborated. For WayDown, the model was run 50 times for German and 91 for Korean stations with the same daily radar input for each station, so that the total length of n-times run was equivalent to 1000 years. Differences between runs are introduced by the model event-values generation process. For LetItRain the model was calibrated to station data and then 1000 years data was simulated as it is a generator model type. |
| Line 445–447/Fig. 13: Are you using the same time period to compare the radar data to RCP2.6 and RCP8.5? | The length of CP data is for both countries 80 years, the length of radar data is 20 / 11 years for Germany and South Korea. We always used the full available length; hence, the differences between extreme statistics between countries could be introduced not only by climate differences, but also due to different data length. We will point it out in conclusion. |
| Line 28: Please remove "of" after "understanding". Line 94–97: I recommend making these two sentences one sentence by stating …"driving variables, and which are not depending…". Line 141: Please remove "i.e." Line 304: You are missing a word between "of" and "means". Line 505: Please remove "the statistics" after "For LetItRain" as this is redundant. | Agreed, will be corrected. |

---

## Author Comment (AC2)

Review 2

| The authors distinguish between "disaggregation models", for which the sum of disaggregated precipitation data is similar or equal to the original coarse precipitation values, and "stochastic models" aiming at reproducing statistics of the original precipitation time series. I do not agree with the terminology used to distinguish the two models (disaggregation versus stochastic), because both models are stochastic, as recognized by the authors at L 259. The main difference is in the statistics the two models preserve from the original dataset. Therefore, I suggest changing the terminology to prevent any misunderstanding. | Agreed, inconsistency of our terminology will be changed. We propose the following way. With the "disaggregation" term we will refer to methods producing time series with increased temporal resolution. "Conditional disaggregation methods/approaches" refer to methods exactly reproducing the value from the reference (daily) dataset ("canonical") or nearly exactly reproducing the value ("microcanonical"). These methods can be of stochastic nature but are not necessarily. "Unconditional disaggregation / rainfall generators" do not reproduce (or only by chance) the reference value and are always stochastic. |
|---|---|
| Validation of both models is done by comparison of simulated and original radar datasets aggregated at the daily scale. In particular, several statistics at the monthly and annual scale for the whole and non-zero time series are used. The authors underline that a straightforward comparison of disaggregated time-series cannot be possible, as the WayDown model only keeps daily precipitation sums consistent with the input data (LL 259-264). This sounds like a major shortcoming of this model compared to the LetItRain model, given the first research question of the study. I suggest to better explain in the conclusions why this model has been chosen rather than other ones available in literature. | Agreed, conclusion will be elaborated. The reasoning behind the model choice was mentioned in the introduction in L91-93, L97-98, namely comparison of two different types of disaggregation models (i.e. pure disaggregation and stochastic generation model). We also mentioned the dissimilar result due to the different characteristics in the two types of models in results (L346-348, 461-462, etc.). We suggest adding the following text in the conclusion. 'In this study, we presented and discussed two different methods to disaggregate the daily output of projected precipitation data to sub-hourly scale. Although both of them belong to a stochastic class of models, the first is a pure disaggregation model which keeps daily sums consistent (WayDown), while the second represents a stochastic rain generator, which mainly focuses on the replication of time-series statistics (LetItRain). Indeed, no studies have undertaken testing of different types of disaggregation models at fine temporal scales, specifically at the 10-minute interval. The outcomes of such a comparative analysis will provide valuable insights into selecting appropriate disaggregation model for urban system analysis.' |
| Concerning the validation in terms of consistency of extreme precipitation frequencies and magnitudes, it is not clear how the 10-min precipitation extremes are | In this case we did not account for event separation, thus both characteristics (extremes under threshold and quantiles) were calculated from the whole time series (both radar and |

| | |
|---|---|
| extracted from the reference and disaggregated datasets. As related to the frequency, the number of 10-min events exceeding given threshold values are calculated and divided by the corresponding number of events in 100 years. How overshooting events in 100 years are calculated? How is it ascertained that selected 10-min extremes are independent from each other (i.e., they don't belong to the same events)? | disaggregated data) as length of time-series with values higher than threshold divided by the whole length. This approach is also commonly met in the literature with regard to disaggregation models validation, along with event-separation and peak-over-threshold extreme analysis (e.g. https://doi.org/10.1038/s41597-022-01304-7, https://doi.org/10.1016/j.jhydrol.2016.07.015). Another reason for not splitting the time-series using i.e. event-based maxima or POT is the limited time-series length (radar observations in S.Korea are 11 years long, thus extreme quantiles estimation for 'reference' data will have even higher uncertainty). A normalisation to 100 years here does not anyhow refer to the return period, rather than to normalise the calculated frequencies for two countries due to different time-series length, thus allowing us to compare the results. We will clarify it in the text. |
| Both models are applied to simulate 5-min precipitation data corresponding to the RCP scenarios 2.6 and 8.5. As far as I understand, WayDown is directly applied to the climate projection datasets, while LetItRain uses the change factor for the mean value and linear regressions between precipitation statistics to obtain parameters for future precipitation generation. In particular linear regressions between the mean and the variance, and between the mean and the proportion of wet periods are used, whereas for high-order moments, i.e., covariance and skewness, historical values are used from the original dataset, as the corresponding linear regressions are not suitable in these cases. Given that the authors state that there are indications that high-order moments of precipitation will change in the future (LL 199-201), I wonder if the authors have tried to apply other techniques for developing nonlinear relationships (e.g., neural networks). If not, a motivation should be provided since, as the authors admit (LL 495-499), when statistics of the current period are preserved and assumed for the future , the application of the disaggregation models on | Agreed, other techniques such as nonlinear regression for developing relationships between high-order moments (skewness) of precipitation could improve the model to better explain the future change in skewness at the fine time scale. Although we have not tried such techniques yet, following the presented reason we want to leave such an investigation for our future study to improve the model if the reviewer accepts. LetItRain was set to reproduce the basic statistics (mean and variance) preferentially. We assigned the lower weight for the covariance (1) and skewness (0.5) than mean (3) and variance (2) in the calibration process. Thus, it doesn't mean that the high-order moments of rainfall used to calibrate the model lead to the same value in the result of the generation. The deviation in autocorrelation between disaggregated and original radar data (Fig. 8) may be found. So before we find the proper relationship between high-order moments of rainfall for future rainfall generation, the model development for reproducing the high-order moments of rainfall should be preceded (L506-508). Therefore, we suggest adding the following text to the conclusion to mention the limitation of the model and its potential improvements. |

| | |
|---|---|
| the climate projection data should be done with caution. | 'Further, the effect of skewness at a fine time scale in the calibration procedure should be investigated more deeply for accurate replication of extreme rainfall. Moreover, a proper relationship between high-order moments of rainfall was not presented. Thus current (historical) statistics for covariance and skewness were directly used as future statistics for model calibration. Nonlinear models such as neural networks could be a possible solution for developing the relationship thus leading to the development of a model that accounts for a change in high-order moments of precipitation characteristics in the future. This will be addressed in the new study.' |
| LL 22-23: Clarify the meaning of "ensemble median" and "ensemble variability" in the abstract. | Agreed, will be added. |
| L 265: "for each station the time-series of 1000-year length equivalent were generated". Do you mean 1000 time series of length equivalent to the observed series or do you generate 1000 years of disaggregated data for each station from which you sample several synthetic datasets of the same length of the observed data? | For WayDown, the model was run 50 times for German and 91 for Korean stations with the same daily radar input for each station, so that the total length of n-times run was equivalent to 1000 years. Differences between runs are introduced by the model event-values generation process. For LetItRain the model was calibrated to station data and then 1000 years data was simulated as it is a generator model type. Will be clarified in methods. |
| LL 276-278: "Corresponding disaggregated events were randomly picked from the models' ensembles …". This sentence is misleading. I suggest replacing "models' ensembles" with generated or synthetic series. | Agreed, will be corrected. |
| Why change factors of the main statistics for the daily scale are represented in Fig. 12 for radar data only? Why the 10-min change factors for radar are not reported? Check the figure legend. | For the Fig.12 Change factors were calculated between RCP and radar data for daily and 10 min scale, thus radar data is included in both cases. In the legend namings were shown accordingly - e.g. 'RCP 2.6 - Radar (daily)'. Here '-' sign means 'between', referring to the change factor between CP and radar data on a daily scale. We suggest changing it to 'and' to avoid confusion (e.g. 'Daily scale: RCP 2.6 and Radar') and to make it more clear for the reader. |